# R-Score: A New Parameter to Assess the Quality of Variants’ Calls Assessed by NGS Using Liquid Biopsies

**DOI:** 10.3390/biology10100954

**Published:** 2021-09-24

**Authors:** Roberto Serna-Blasco, Estela Sánchez-Herrero, María Berrocal Renedo, Silvia Calabuig-Fariñas, Miguel Ángel Molina-Vila, Mariano Provencio, Atocha Romero

**Affiliations:** 1Liquid Biopsy Laboratory, University Hospital Puerta de Hierro, 28222 Madrid, Spain; rserna@idiphim.org (R.S.-B.); esanchez@idiphim.org (E.S.-H.); mariaberrocal18@gmail.com (M.B.R.); 2Atrys Health, I+D Department, 08025 Barcelona, Spain; 3CIBERONC, Liquid Biopsy WM, 28029 Madrid, Spain; calabuix_sil@gva.es; 4Mixed Unit TRIAL, Príncipe Felipe Research Center & General University Hospital of Valencia Research Foundation, 46012 Valencia, Spain; 5Department of Pathology, Universitat de València, 46010 Valencia, Spain; 6Laboratory of Oncology/Pangaea Oncology, Quirón-Dexeus University Hospital, 08028 Barcelona, Spain; mamolina@panoncology.com; 7Medical Oncology, University Hospital Puerta de Hierro, 28222 Madrid, Spain; mariano.provencio@salud.madrid.org

**Keywords:** NGS, ctDNA, VAF, liquid biopsy, filtering, variant calling

## Abstract

**Simple Summary:**

Circulating tumor DNA profiling by next-generation sequencing (NGS) is becoming essential for guiding targeted therapies. However, it remains challenging. Here, we show that variant allele fraction and the median of absolute values of all pairwise differences impact the agreement between digital PCR and NGS calls. Therefore, we propose a new parameter, named R-score, which integrates both variables, and we evaluate its usefulness for optimizing NGS variant calling.

**Abstract:**

Next-generation sequencing (NGS) has enabled a deeper knowledge of the molecular landscape in non-small cell lung cancer (NSCLC), identifying a growing number of targetable molecular alterations in key genes. However, NGS profiling of liquid biopsies risk for false positive and false negative calls and parameters assessing the quality of NGS calls remains lacking. In this study, we have evaluated the positive percent agreement (PPA) between NGS and digital PCR calls when assessing *EGFR* mutation status using 85 plasma samples from 82 *EGFR*-positive NSCLC patients. According to our data, variant allele fraction (VAF) was significantly lower in discordant calls and the median of the absolute values of all pairwise differences (MAPD) was significantly higher in discordant calls (*p* < 0.001 in both cases). Based on these results, we propose a new parameter that integrates both variables, named R-score. Next, we sought to evaluate the PPA for *EGFR* mutation calls between two independent NGS platforms using a subset of 40 samples from the same cohort. Remarkably, there was a significant linear correlation between the PPA and the R-score (r = 0.97; *p* < 0.001). Specifically, the PPA of samples with an R-score ≤ −1.25 was 95.83%, whereas PPA falls to 81.63% in samples with R-score ≤ 0.25. In conclusion, R-score significantly correlates with PPA and can assist laboratory medicine specialists and data scientists to select reliable variants detected by NGS.

## 1. Introduction

The analysis of circulating tumor DNA (ctDNA) has become an attractive approach for non-invasive biomarker testing as well as for real-time monitoring of cancer patients; its usefulness is especially remarkable in lung cancer patients [1,2,3,4]. These tumors are mostly diagnosed at advanced stages, in elderly patients with a median age at diagnosis of approximately 65 years [5], and they are difficult to access owing to their anatomical location, which makes it sometimes difficult to obtain sufficient material for molecular analysis [6]. Moreover, in the last decades, there has been a major paradigm shift in the management of metastatic non-small cell lung cancer (NSCLC), with the advent of targeted therapies for patients harbouring druggable alterations such as *EGFR* or *BRAF* mutations, as well as *ALK*, *ROS*, and *RET* rearrangements, and so on [7]. Furthermore, novel *KRAS* inhibitors constitute a promising therapeutic approach for advanced NSCLC patients [8,9]. Specifically, 30% of NSCLC tumours harbour activating mutations in the *EGFR* gene, which identify patient candidates to receive tyrosine kinase inhibitors (TKIs) [10]. For this subset of patients, liquid biopsy has been proven to be extremely useful, saving time in the process of diagnosis. In this way, guidelines recommend testing for the T790M *EGFR* mutation in the blood after progression to an *EGFR* TKI as a first choice, and re-biopsies are suggested in the case of a negative result in order to identify patients that can benefit from osimertinib (a third-generation TKI) [11]. Moreover, ctDNA plasma levels have been shown to be of prognostic significance for these patients, and monitoring *EGFR* mutation levels in the plasmas has been proven useful for response to treatment monitoring [5,12,13].

Next-generation sequencing (NGS) enables simultaneous detection of multiple alterations in a single test. Incorporation of unique molecular identifiers (UMIs), random nucleotide sequences assigned to each DNA fragment prior to PCR amplification during library preparation, enables the detection, quantification, and sequencing of unique DNA fragments with high-resolution, allowing the identification and removal of amplification artifacts arising from library preparation and the reduction of the limit of detection (LOD) [14,15]. Nonetheless, ctDNA is present at very low levels in the plasma and its profiling is still challenging with working conditions sometimes close to the edge of this technology. Therefore, there is a need to develop new parameters assessing the quality of the reads in order to avoid false positive and false negative calls.

Here, we assess the impact of two parameters, namely, variant allele fraction (VAF) and median of the absolute values of all pairwise differences (MAPD), separately and together on variant calls when using the Oncomine Pan-Cancer Cell-Free Assay™ (ThermoFisher Scientific^®^, Palo Alto, CA, USA) by evaluating the agreement between digital PCR (dPCR) and NGS for the assessment of *EGFR* mutation status. Based on our data, we propose a new parameter named R-score and, finally, we evaluate the agreement in NGS calls between two independent NGS methods according to R-score.

## 2. Materials and Methods

### 2.1. Patients and Samples

A total of 85 samples from advanced *EGFR*-positive NSCLC patients were recruited upon disease progression to a first-line with a TKI, between February 2016 and March 2019. The study was approved by the Hospital Puerta de Hierro Ethics Committee. All patients provided the appropriate written informed consent to participate in the study prior to enrolment. Briefly, eligible patients were both male and female, age >18 years, with a pathologically confirmed diagnosis of stage IV NSCLC harbouring an *EGFR* mutation. Blood samples were collected in 8.5 mL PPT^TM^ tubes (Becton Dickinson, Franklin Lakes, NJ, USA).

### 2.2. Laboratory Procedures

Two independent laboratories were involved in this study: laboratory 1 (L1) and laboratory 2 (L2). Samples for which we did not have available at least 8 mL of plasma (N = 45) were processed only by L1 exclusively, and they were used to test the agreement between dPCR and NGS exclusively. For 40 plasma samples, we had available at least 8 mL of plasma, and samples were divided into two aliquots, which were then distributed to L1 and L2. L1 carried out dPCR assays and NGS analysis using the Oncomine Pan-Cancer Cell-Free Assay and an Ion S5 sequencer (ThermoFisher Scientific^®^, Palo Alto, CA, USA), whereas L2 carried out NGS with QIAact Lung DNA UMI Panel using the GeneRead Platform (QIAgen, Valencia, CA, USA).

Isolation of plasma was achieved by two consecutive centrifugations at room temperature, the first one at 1500× *g* for 10 min and the second at 5000× *g* for 20 min. cfDNA was extracted with the QIAamp Circulating Nucleic Acid Kit (QIAgen, Valencia, CA, USA) according to the manufacturer’s protocol (QIAamp Circulating Nucelic Acid Handbook 10/2013). DNA concentration was measured by Qubit 2.0 Fluorometer with Qubit 1X dsDNA HS Assay Kit (ThermoFisher Scientific^®^, Palo Alto, CA, USA) and fragment length and sample quality were evaluated using the Agilent High Sensitivity DNA Kit using Agilent 2100 Bioanalyzer (Agilent Technologies, Santa Clara, CA, USA). Appendix A shows the observed size of the cfDNA fragments, which was approximately 180 bp. cfDNA was stored at −80 °C until further analysis.

In order to detect somatic mutation in the *EGFR* gene, dPCRs were performed using predesigned TaqMan^®^ dPCR assays in a QuantStudio^®^ 3D Digital PCR (Applied Biosystems^®^, South San Francisco, CA, USA). dPCR reaction was carried out in a final volume of 18 µL; this reaction included 8.55 µL of template cfDNA, 9 µL of 20X QuantStudio^®^ Master Mix, and 0.45 µL 40X TaqMan assay. Subsequently, 14.5 µL of final reaction volume was loaded to QuantStudio^®^ 3D digital PCR 20K chip. The thermal cycler conditions were as follows: initial denaturalization at 96 °C for 10 min, 40 cycles at 56 °C for 2 min, 98 °C for 30 s, 72 °C for 10 min, and finally samples were maintained at 22 °C for at least 30 min. Chips were read using QuantStudio^®^ 3D Digital PCR instrument. The results were analysed with QuantStudio^®^ 3D AnalysisSuite™ Cloud. Default call assignments for each data cluster were manually adjusted when needed. A positive and a negative control were included in every run. The LOD and limit of quantitation of the dPCR TaqMan^®^ assays were estimated based on the standard deviation of the response and the slope according to the recommendations of The International Council for Harmonisation of Technical Requirements for Pharmaceuticals for Human Use; ICH Q2 (R1) guidelines (validation of analytical procedures: text and methodology), and they have been published elsewhere [13]. The sensitivity and specificity of the assays, considering tissue genotyping to be the gold standard, have also been reported [16].

The presence of *EGFR* mutations was evaluated in parallel by two independent NGS platforms, Ion S5™ XL and GeneReader™, and using two different gene panels, Oncomine™ Pan-Cancer Cell-Free Assay (ThermoFisher Scientific^®^, Palo Alto, CA, USA) and the QIAact Lung DNA UMI Panel (QIAgen, Valencia, CA, USA), respectively. The comparison was performed using 40 samples.

For NGS analysis using the Oncomine Pan-Cancer Cell-Free Assay (NGS-Oncomine), library preparation was performed with a minimum input of 10 ng of cfDNA according to manufacturer’s instructions. The final pool was loaded in an Ion 550™ Chip using Ion Chef™ Instrument (ThermoFisher Scientific^®^, Palo Alto, CA, USA). Finally, loaded chips were sequenced on an Ion GeneStudio™ S5 Sequencer (ThermoFisher Scientific^®^, Palo Alto, CA, USA). Torrent Suite Software (v5.12) was used to perform raw sequencing data analysis. The CoverageAnalysis (v. 5.12.0.0) plugin was used for sequencing coverage analysis (ThermoFisher Scientific^®^, Palo Alto, CA, USA). As recommended by the manufacturer, a median read coverage > 25,000 and median molecular coverage > 2500 were required to detect a variant with a VAF of 0.1%. Raw reads were aligned to the human reference genome hg19. Variant calling, annotation, and filtering were performed on the Ion Reporter (v5.10) platform using the OncomineTagSeq Pan-Cancer Liquid Biopsy workflow (v2.1). Briefly, sequencing reads were mapped to defined target regions (Oncomine Pan-Cancer DNA Regions v1.0 (5.10)) and subjected to variant calling using Oncomine Pan-Cancer Annotations v1r.0.

For NGS analysis using the QIAact Lung DNA UMI Panel (NGS-GeneReader), libraries were performed with an input of 16.75 μL and ~10–70 ng of purified cfDNA, according to manufacturer’s instructions. Then, libraries were quantified using a QIAxcel Advanced System and Qubit dsDNA HS Assay kit in order to pool in batches of six samples. GeneRead Clonal Amp Q Kit was used to clonal amplification of pooled libraries. After bead enrichment, pooled libraries were sequenced using the GeneRead UMI Advanced Sequencing Q kit in a GeneReader instrument. Finally, FASTQ files alignment was performed using hg19 as reference genome, and variant calling and report generation of sequencing results were performed by QIAGEN Clinical Insight Analyze software.

### 2.3. Parameters

VAF was defined as the number of mutant molecules at a specific nucleotide location relative to the sum of total DNA molecules (mutant + wild type). VAF was provided for each detected mutation after dPCR and NGS analysis. In dPCR analysis, VAF was calculated, following the next equation, by QuantStudio^®^ 3D AnalysisSuite™ Cloud:VAF = (FAMcopies/µL)/(FAMcopies/µL + VICcopies/µL) × 100(1)
where FAM copies = number of reads of mutated sequences and VIC copies = number of reads of wild-type sequences.

In the case of NGS-Oncomine, VAF was calculated, using the CoverageAnalysis (v. 5.12.0.0) plugin. Likewise, using NGS-GeneReader, VAF was calculated with QIAGEN Clinical Insight Analyze software in the same way as NGS-Oncomine.

NGS-Oncomine platform also provides a quality sequencing parameter, MAPD, as a pair is defined as adjacent amplicons in terms of genomic distances. Assuming that adjacent amplicons in the genome most likely have the same underlying copy number in a sample, the difference between the log2c(read count ratio) values against the reference baseline for all adjacent amplicons contains information for the noise level of the data. The MAPD is an estimation of coverage variability between adjacent amplicons. The default threshold is 0.5 [17]. As a result, sample results with an MAPD above this value should be reviewed with caution
MAPD = median(|x_i+1_−x_i_|)(2)
where x_i_ = log2 ratio for marker i.

### 2.4. Statistical Analysis

The primary objective was to evaluate the impact of VAF and MAPD parameters, separately and together, firstly on the positive percent agreement (PPA) between dPCR and NGS (NGS-Oncomine) and secondly on the PPA between two independent NGS platforms (NGS-Oncomine and NGS-GeneReader).

Each mutation was treated as a separate measurement for statistical analysis; therefore, 137 measurements were used in this study.

The correlation between VAFs measured by dPCR and NGS was assessed with simple linear regression analysis, using the concordance correlation coefficient (p) and Spearman’s coefficient (r). For comparisons between numerical variables, Mann–Whitney U test was used. Comparisons between categorical variables were made using Fisher’s exact test or chi-squared test, whichever was most appropriate.

To describe how often NGS and dPCR methods agreed on *EGFR* calls, as well as concordance between the two different NGS platforms, we calculated the PPA.

The threshold of *p* < 0.05 was considered as statistically significant. Statistical software used was Stata v16.0 (StataCorp 2019. Stata Statistical Software Release 16. College station, TX: StataCorp LLC) and R version 3.6.3. (R core team 2020. The R Foundation for Statistical Computing Platform, Vienna, Austria) URL https://www.R-project.org/ (last accessed on 26 July 2021).

## 3. Results

### 3.1. EGFR Mutation Detection by dPCR and NGS-Oncomine

EGFR mutation status was evaluated in 85 plasma samples from 82 *EGFR*-positive NSCLC patients in parallel by dPCR and NGS-Oncomine. All samples used in this study had detectable *EGFR* driver mutations by dPCR. The mutation detected by dPCR was always concordant with the *EGFR* mutation detected on the pre-treatment tissue sample as reported by pathologists. Among the total number of detected *EGFR* mutations (N = 137), 62% were activating *EGFR* mutations, among which the most common mutations were exon 19 deletions (55.3%) or L858R (36.5%). The rest of the *EGFR* driver mutations were L861Q (3.5%), G719A (2.3%), S768I (1.2%), and exon 20 insertions (1.2%). Regarding T790M resistance mutation, 61.2% of samples were identified as T790M positive by dPCR. Data for T790M status in tissue samples were not available. Of note, 42 (30.6%) mutations detected by dPCR were not found using NGS-Oncomine. When analysing mutations separately, a lower PPA was measured in L858R mutation (67.74%; 95%CI 50.31–85.17) compared with exon 19 deletion (76.60%; 95%CI 64.03–89.16). Less common *EGFR* mutations such as L861Q, S768I, and G719A were detected by both methods. It should be noted that the exon 20 insertion (c.2310_2311insGGT; p.D770_N771insG) was not found by NGS-Oncomine.

Finally, regarding T790M resistance mutation, 52 (61.2%) samples were identified as T790M positive by dPCR, whereas only 28 (33%) samples were T790M positive using NGS-Oncomine (53.85% of agreement; 95% CI 39.83–67.86).

### 3.2. VAF and MAPD Involvement in the Agreement between dPCR and NGS-Oncomine Calls

Overall, there were 91 concordant calls by both technologies and 46 discordant calls with a PPA of 66.42% (95% CI 58.42–74.43).

First, we evaluated the overall correlation between VAF values assessed by dPCR and NGS-Oncomine when the mutation was detected by both methods. According to our data, VAFs measured by NGS-Oncomine were significantly correlated to VAFs assessed by dPCR (r = 0.89; *p* < 0.001) (Appendix A). Next, VAFs values and MAPD scores were compared between discordant and concordant calls. Overall, dPCR VAFs values were significantly lower in discordant calls compared with concordant calls (*p* < 0.001) (Figure 1A). Specifically, 1.1% and 10.9% of concordant calls have VAF ≤ 0.1% and ≤ 0.5%, respectively, compared with 8.9% and 46.7% in discordant calls. Likewise, MAPD score was significantly higher in discordant samples compared with concordant samples (*p* < 0.001) (Figure 1B).

Next, we sought to evaluate the combined effect of VAF and MAPD parameters. Dot plots in Figure 2 show the concordance between dPCR and NGS-Oncomine on variant calls according to VAF and MAPD parameters. Discordant calls are coloured in red and concordant calls are coloured in blue. Figure 2A is divided into four quadrants using as cut-offs the logarithmic median values of VAF and MAPD according to our data set. As shown, the highest PPA (96.9%; 95%CI: 83.8–99.9%) was observed in the lower-right quadrant. Conversely, the PPA descended as much as 27.6% (95%CI: 12.7–47.2%) for calls clustered in the upper-left quadrant, meaning that, the higher the VAF and the lower the MAPD, the higher the PPA. Similar results were obtained when quadrants were divided using thresholds according to technical specifications for each parameter (Figure 2B). As illustrated, PPA between NGS and dPCR calls was 0% (95% CI: 0–60.2%) when using a cut off of ≤−0.301 for VAF and >−0.301 for MAPD, whereas in the opposite conditions, the PPA increased to 84.9% (95% CI: 74.5–90.9%).

### 3.3. R-Score Is a Useful Parameter to Select Reliable Variant Calls

Based on previous observations, we proposed a new parameter, named R-score, which is defined as follows:R-score = log(MAPD/VAF)(3)

In order to evaluate the utility of R-score for assessing the quality of an *EGFR* variant call, we evaluated the PPA between NGS-Oncomine and dPCR and NGS-Oncomine and NGS-GeneReader according to R-score.

First, we assessed the correlation between VAF values from NGS-Oncomine and NGS-GeneReader when the mutation was detected by both methods. According to our data, VAFs values from NGS-Oncomine significantly correlated with VAFs from NGS-GeneReader (r = 0.80; *p* < 0.001).

R-score was then calculated for each variant detected by NGS-Oncomine using the VAF and MAPD provided by the corresponding analysis software. MAPD and R-score values were significantly higher in discordant calls between dPCR and NGS-Oncomine compared with concordant calls (*p* < 0.001) (Figure 3A and Appendix A). Conversely, VAF values were significantly lower in discordant calls between dPCR and NGS-Oncomine (Appendix A). Subsequently, the PPA for *EGFR* variant calling between both NGS platforms was evaluated using different arbitrary R-score cut-offs (−1.25, −1, −0.75, −0.5, −0.25, 0, and 0.25). As shown in Figure 3C, there was a clear linear correlation between the PPA and the R-score (r = 0.97; *p* < 0.001). Of note, the PPA of samples with an R-score ≤ −1.25 was 95.83%, whereas PPA falls to 81.63% in samples with an R-score ≤ 0.25 (Figure 3B). A complete list of all mutations detected according to the NGS platform is available in Appendix A.

## 4. Discussion

Biomarker testing in NSCLC has been demonstrated to improve survival outcomes [19,20,21]. Of note, the number of biomarkers that need to be tested is constantly increasing in NSCLC as new targeted therapies are becoming available [7]. Unlike PCR-based platforms, which only allow a few mutations to be analyzed, NGS enables for interrogating multiple genomic alterations simultaneously in a single test. Indeed, National Comprehensive Cancer Network guidelines recommend that, when feasible, biomarker testing should be performed via a broad, panel-based approach by NGS [2]. However, NGS profiling of liquid biopsies, although feasible [22,23], remains challenging. On one hand, the sensitivity of the assays remains a major limitation [24], and approaches aimed to increase sensitivity might risk false positive calls. Moreover, it has been reported that tumor mutational burden (TMB) analysis, which has been proposed as a predictive biomarker for the identification of patients most likely to respond to immune checkpoint inhibitors, through liquid biopsies, is feasible [25]. TMB is optimally assessed by whole-exome sequencing (WES) [26], but targeted panels provide a time-effective and cost-effective alternative [27]. Nevertheless, TMB analysis requires sequencing over 0.5 Mb [28,29]. In this scenario, it is of particular interest to reduce as much as possible the risk of false-positive and false-negative calls. Thus, new parameters evaluating the quality of NGS calls are needed. A recent comprehensive study, in which several methodologies for the analysis of circulating tumor DNA were compared, revealed that the agreement between platforms significantly improved when discarding samples with VAF ≤ 0.5% [16]. Likewise, a study comparing BEAMing and droplet dPCR for ctDNA analysis using plasma samples from advanced breast cancer patients enrolled in the PALOMA-3 trial showed that discordant calls occurred at VAFs < 1% [30]. In the view of our findings, we hypothesized that the combination of VAF with MAPD could further improve the assessment of the reliability of a variant call. According to our data, MAPD was significantly higher in discordant samples compared with concordant calls (*p* < 0.001), while VAF values were significantly lower in discordant calls compared with concordant samples (*p* < 0.001). Remarkably, as shown in Figure 2, the highest PPA (96.9%; 95%CI: 83.8–99.9%) was observed in the lower-right quadrant. Conversely, the PPA descended as much as 27.6% (95%CI: 12.7–47.2%) for calls clustered in the upper-left quadrant.

Our results are limited to *EGFR* locus as the cohort included exclusively *EGFR*-positive NSCLC patients. Nonetheless, mutations in other key genes were found. Specifically, in our data set, there were two samples testing positive for *KRAS* mutations by both NGS platforms (data not shown). Larger cohorts assessing the utility R-score for assessing the reliability of variant calls in other loci different may be of particular interest.

Taken together, we propose the R-score defined as the log(MAPD/VAF). According to our results, *EGFR* variants with positive R-score are particularly sensitive to genotyping errors. As presented in Figure 3, a significant correlation was found between PPA and the R-score cut-off values, indicating that R-sore can be useful to discriminate between true and false calls in the *EGFR* locus.

## 5. Conclusions

VAF and MAPD have an impact on *EGFR* variant calling. Combining this information in a score (R-score) can further improve the assessment of the reliability of a variant call. Using a dataset of 85 *EGFR*-positive NSCLC patients, we find that *EGFR* variants with positive R-score are particularly sensitive to erroneous variant calls in the *EGFR* gene.

## Figures and Tables

**Figure 1 biology-10-00954-f001:**
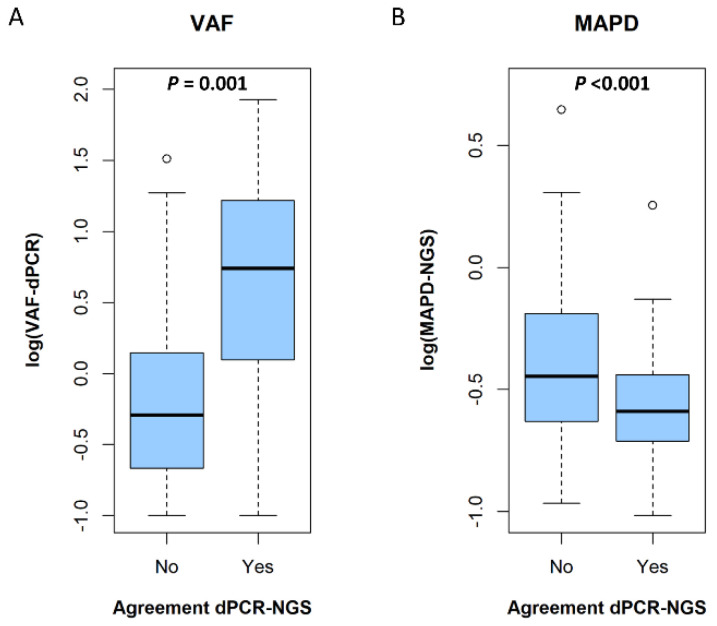
Boxplot. (**A**) VAF values in concordant and discordant calls (dPCR-NGS-Oncomine) in logarithmic scale. (**B**) MAPD values in concordant and discordant calls (dPCR-NGS-Oncomine) in logarithmic scale.

**Figure 2 biology-10-00954-f002:**
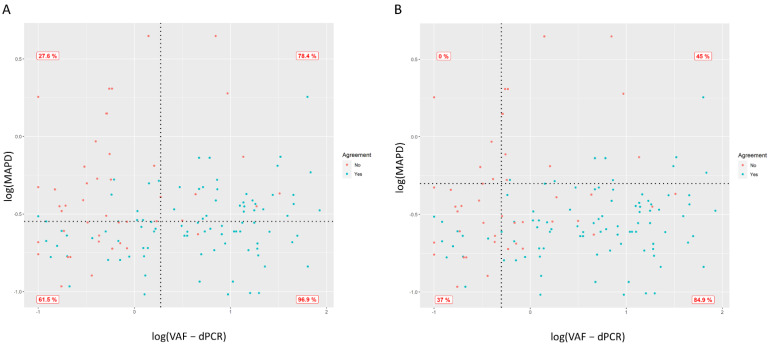
Dot plot showing the agreement in variant calls between dPCR—NGS-Oncomine, according to VAF and MAPD. VAF and MAPD values, both in logarithmic scale, are represented in the *x* and *y*-axis, respectively. Concordant calls are coloured in blue, while discordant calls are coloured in red. PPA for calls clustered in each quadrant is shown. (**A**) Dot plot divided into four quadrants using as cut-off the logarithmic median values of VAF and MAPD. In this way, the median VAF in our data set was 1.87, which is 0.272 on logarithmic scale, and the median MAPD was 0.28, which corresponds to −0.553 on logarithmic scale. (**B**) Dot plot divided into four quadrants according to technical specifications. MAPD threshold was selected following Ion Reporter recommendations [17]. According to the manufacturer, a value of MAPD above 0.5 is considered too high. Samples with high MAPD values have low coverage uniformity, which can result in missed or erroneous variant calls. The VAF threshold was chosen based on results from previous studies [18]. Therefore, both axes were divided using −0.301 value for VAF and MAPD (log(0.5)).

**Figure 3 biology-10-00954-f003:**
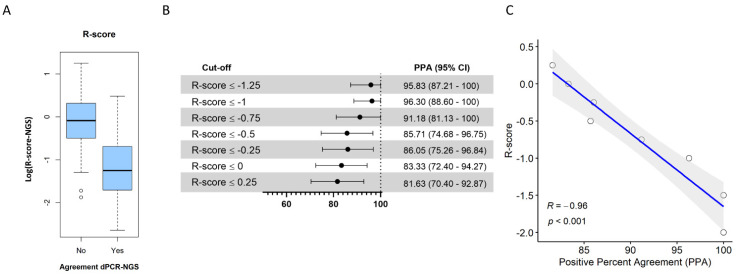
(**A**) R-score values in logarithmic scale in concordant and discordant samples (dPCR—NGS-Oncomine). R-score was calculated for each variant detected with Oncomine panel, using VAF and MAPD values from Ion Reporter (v5.10) analysis software. (**B**) Positive percent agreement (PPA) with corresponding 95% confidence interval (95%CI) between NGS-Oncomine and NGS-GeneReader according to R-score cut-off. The following arbitrary cut-offs for R-score were established: −1.25, −1, −0.75, −0.5, −0.25, 0, and 0.25, and PPA between both NGS platforms was estimated. (**C**) Correlation between PPA values and R-score cut-off values. As shown, there was a linear correlation between PPA and the R-score cut-off values; the lower the R-score, the greater the PPA. Abbreviations: R = Spearman correlation coefficient.

## Data Availability

All data generated or analyzed during this study are included in this published article (and its Appendix A).

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
