# Peer review of "R-Score: A New Parameter to Assess the Quality of Variants’ Calls Assessed by NGS Using Liquid Biopsies"

_biology, 2021, doi:10.3390/biology10100954_

Round 1

Reviewer 1 Report

The authors have addressed the minor issues listed but did not address the fundamental weakness of the manuscript.

  1. The lack of rigorous verification in cohorts of NSCLC patients with and “without" the EGFR mutations makes it not so useful to scientists. KRAS mutation occurs in approximately 25% of NSCLC patients and can also be detected by both Oncomine and GeneReader panels. Whether R-score is also applicable on KRAS mutation detection. This issue is important to evaluate the false positive/false negative mutation detection by R-score.
  2. Wrong labeling: Figure S1A indicates two sample electropherogram and S1B indicates a gel-like image.
  3. Line 343: Figure S2 needs to relocate before Table S1
  4. Table S2 may be present in an excel file as supplementary data and include R-score values. (Typo: MAF?)

Author Response

REVIEWER 1

The authors have addressed the minor issues listed but did not address the fundamental weakness of the manuscript.

The lack of rigorous verification in cohorts of NSCLC patients with and “without" the EGFR mutations makes it not so useful to scientists. KRAS mutation occurs in approximately 25% of NSCLC patients and can also be detected by both Oncomine and GeneReader panels. Whether R-score is also applicable on KRAS mutation detection. This issue is important to evaluate the false positive/false negative mutation detection by R-score.

Please note that this cohort corresponds to EGFR-positive NSCLC patients. It is well established that concomitant KRAS and EGFR mutations are unusual as EGFR, KRAS, and ALK mutations are widely known as mutually exclusive.  According to our data, KRAS mutations were detected by both panels in two samples. Please see the table in the attached file.

Unfortunately, we don’t have more KRAS mutations in our data set and it is not possible to increase the sample size within 10 days since we have received the Major Revisions notification from the journal.

Please note that we specify in the revised version that results are restricted to EGFR locus in the discussion section:

According to our results, EGFR variants with positive R-score are particularly sensitive to genotyping errors. As presented in Figure 3, a significant correlation was found between PPA and the R-score cut-off values, indicating that R-sore can be useful to discriminate between true and false calls in the EGFR locus.

Wrong labeling: Figure S1A indicates two sample electropherogram and S1B indicates a gel-like image.

Please note that we have corrected figure S1 caption.

Figure S1. (A) Two samples electropherogram showing 180 bp peak corresponding to low fragment cfDNA and peaks corresponding to mono-, di- and tri-nucleosomes. (B) Gel-like image of cfDNA samples analysed with Bioanalyzer 2100

Line 343: Figure S2 needs to relocate before Table S1.

Please note that Figure S2 has been relocated.

Table S2 may be present in an excel file as supplementary data and include R-score values. (Typo: MAF?)

Please note that Table S2 is now also provided as an excel file in order to improve its visualization and the R-score column has been added at the right side of this table. MAF has been replaced by VAF in the Table S2 column titles.

Table S2 is also provided as an excel file. (Table S2.xlsx).

Reviewer 2 Report

Comments:

  1. Line 33: The authors mention: “… using a subset of 70 samples from the same cohort”. This I find little confusing; hence, the authors also mention the use of 85 plasma samples in line 28 and 90. Can the authors lay the fundamentals of how there is a difference of 15 samples?
  2. Section 3.1. The authors nicely present their results of concordances between dPCR and NGS, and from the supplementary figure 2 there is a strong correlation between VAFs between the two methods. In Line 219-220, the authors mention that “…(c.2310_2311insGGT; p.D770_N771insG) was not found by NGS-Oncomine”, it could be interesting to know briefly if the variant was not called by the Ion Suite Software v.5.12 software or if the insertion was absent completely in raw data. The latter can be achieved via for instance integrative genomics viewer or similar.
  3. The authors rightfully use the variant allele frequency (VAF) throughout the paper. However, in the supplementary table 2 containing all identified EGFR mutations the authors use minor allele frequency (MAF) I would assume (MAF not defined). I would suggest you define what you mean specifically by MAF. From my view, Minor allele frequency refers to an allele frequency of which is observed in a population of patients. Whereas, I find the VAF used in a clinical setting to describe the percentage of reads of a variant allele for a single patient.
  4. Although you wrap up the study in the discussion section, I would suggest to make a clear concise conclusion as a final section.       

Minor comments:

  1. This applies to all 3 sections of the manuscript; it is advised that abbreviations are made if the abbreviated words are used in the text.
    1. Simple summary: Next generation sequencing is abbreviated NGS and used later in the text. ctDNA, VAF, dPCR, MAPD abbreviations of words not used further in the simple summary.
    2. Abstract: NGS, NSCLC, PPA abbreviations for words used in the text. VAF not further used. dPCR not defined.
    3. Main text: Words should only be abbreviated once and then consistently used. LOD defined in line: 76 & 127/128, MAPD defined in line: 82 & 174/175. VAF defined in line: 81 & 163. PPA defined in line 186 & 197/198. LOQ defined in line: 128, abbreviation not used in the text. NCCN in line: 305, not defined.
  2. The nomenclature of human genes is in capital and in italics.  
  3. Line 111: “…manufacture’s protocol” – Add which protocol this is.
  4. Table S2: I would suggest adding a grid to make it more eye friendly to follow. If possible, choose a font size so that the text can fit in one line horizontally.

Author Response

Line 33: The authors mention: “… using a subset of 70 samples from the same cohort”. This I find little confusing; hence, the authors also mention the use of 85 plasma samples in line 28 and 90. Can the authors lay the fundamentals of how there is a difference of 15 samples?

We would like to acknowledge the reviewer for this observation. In the abstract we mentioned  “we sought to evaluate the PPA for EGFR mutation calls between two independent NGS platforms using a subset of 70 samples from the same cohort” and the correct statement would be “we sought to evaluate the PPA for EGFR mutation calls between two independent NGS platforms using a subset of 40 samples from the same cohort”. Please note that this number has been corrected in the abstract section.

Please note that as explained in the MM section 40 plasma samples were used for the evaluation of  PPA between NGS-Oncomine and NGS-GeneReader according to R-score

Two independent laboratories were involved in this study: Laboratory 1 (L1) and laboratory 2 (L2). Samples for which we did not have available at least 8 mL of plasma (N=45) were processed only by L1 exclusively and they were used to test the agreement between dPCR and NGS exclusively. For 40 plasma samples, we had available at least 8 mL of plasma, and samples were divided into two aliquots which were then distributed to L1 and L2. L1 carried out dPCR assays and NGS analysis using the Oncomine Pan-Cancer Cell-Free Assay and an Ion S5 sequencer (Thermo Fisher, Palo Alto, CA), whereas L2 carried out NGS with QIAact Lung DNA UMI Panel using the GeneReader Platform (QIAgen, Valencia, CA, USA)

Section 3.1. The authors nicely present their results of concordances between dPCR and NGS, and from the supplementary figure 2 there is a strong correlation between VAFs between the two methods. In Line 219-220, the authors mention that “…(c.2310_2311insGGT; p.D770_N771insG) was not found by NGS-Oncomine”, it could be interesting to know briefly if the variant was not called by the Ion Suite Software v.5.12 software or if the insertion was absent completely in raw data. The latter can be achieved via for instance integrative genomics viewer or similar.

We thank the reviewer for this commentary. First, we have checked the BED file which describes target regions (target_region_Oncomine_Pan-Cancer_DNA_Regions_v1.1.bed), used by the workflow that was used for the analysis of these samples (OncomineTagSeq Pan-Cancer Liquid Biopsy workflow (v2.1)). As hypothesized by the reviewer, the amplicon that contains the genomic region of the variant c.2310_2311insGGT; p.D770_N771insG; COSM12378, is configured to assess the following variants; COSM12377, COSM12376, COSM13428, COSM12381, and COSM6241 in the BAM file. Therefore, our variant of interest cannot be called by the Ion Suite Software v.5.12 software.

However, we also checked the BAM files from this sample using Integrative Genomics Viewer. As we show in the IGV screenshot (please refer to the attached file) this insertion does not appear in the raw data.

The authors rightfully use the variant allele frequency (VAF) throughout the paper. However, in the supplementary table 2 containing all identified EGFR mutations the authors use minor allele frequency (MAF) I would assume (MAF not defined). I would suggest you define what you mean specifically by MAF. From my view, Minor allele frequency refers to an allele frequency of which is observed in a population of patients. Whereas, I find the VAF used in a clinical setting to describe the percentage of reads of a variant allele for a single patient.

Please note that MAF has been replaced by VAF in the Table S2 column titles.

Although you wrap up the study in the discussion section, I would suggest to make a clear concise conclusion as a final section.      

Please note that we have included a new conclusion section.

  1. Conclusions

VAF and MAPD have an impact on EGFR variant calling. Combining this information in a score (R-score) can further improve the assessment of the reliability of a variant call. Using a dataset of 85 EGFR-positive NSCLC patients we EGFR variants with positive R-score are particularly sensitive to erroneous variant calls in EGFR gene.

Minor comments:

This applies to all 3 sections of the manuscript; it is advised that abbreviations are made if the abbreviated words are used in the text.

Simple summary: Next generation sequencing is abbreviated NGS and used later in the text. ctDNA, VAF, dPCR, MAPD abbreviations of words not used further in the simple summary.

Abstract: NGS, NSCLC, PPA abbreviations for words used in the text. VAF not further used. dPCR not defined.

Main text: Words should only be abbreviated once and then consistently used. LOD defined in line: 76 & 127/128, MAPD defined in line: 82 & 174/175. VAF defined in line: 81 & 163. PPA defined in line 186 & 197/198. LOQ defined in line: 128, abbreviation not used in the text. NCCN in line: 305, not defined.

Following the reviewer’s advice, we have corrected all issues related to abbreviations in the main text.

The nomenclature of human genes is in capital and in italics. 

Please note that all human genes are now in italics.

Line 111: “…manufacture’s protocol” – Add which protocol this is.

Please note that we have added the manufacturer's protocol

According to the manufacture’s protocol (QIAamp Circulating Nucelic Acid Handbook 10/2013).

Table S2: I would suggest adding a grid to make it more eye friendly to follow. If possible, choose a font size so that the text can fit in one line horizontally.

Please note that Table S2 is now also provided as an excel file in order to improve its visualization and the R-score column has been added at the right side of this table. MAF has been replaced by VAF in the Table S2 column titles.

Table S2 is also provided as an excel file. (Table S2.xlsx).

Round 2

Reviewer 1 Report

It is pity that the limited significance of R-score in the assessing mutation (only for EGFR variants).

Author Response

Please note that we mention the limitation of the study in the Discussion Section. 

Our results are limited to the EGFR locus as the cohort included exclusively EGFR-positive NSCLC patients. Nonetheless, mutations in other key genes were found. Specifically, in our data set there were two samples testing positive for KRAS mutations by both NGS platforms (data not shown). Larger cohorts assessing the utility R-score for assessing the reliability of variant calls in other loci different may be of particular interest.